# Tunable Young’s Moduli of Soft Composites Fabricated from Magnetorheological Materials Containing Microsized Iron Particles

**DOI:** 10.3390/ma13153378

**Published:** 2020-07-30

**Authors:** Ji-Young Yoon, Seong-Woo Hong, Yu-Jin Park, Seong-Hwan Kim, Gi-Woo Kim, Seung-Bok Choi

**Affiliations:** Smart Structure and Systems Laboratory, Department of Mechanical Engineering, Inha University, Incheon 22212, Korea; ji_young62@naver.com (J.-Y.Y.); ghdtjddn3@gmail.com (S.-W.H.); pyjn5059@naver.com (Y.-J.P.); neumann9177@naver.com (S.-H.K.)

**Keywords:** soft composite, skin layer, magnetorheological elastomer, magnetorheological fluid, tensile test, equivalent Young’s modulus

## Abstract

This study experimentally investigates the field-dependent Young’s moduli of soft composites, which are fabricated from two different magnetic-responsive materials; magnetorheological elastomer (MRE) and magnetorheological fluid (MRF). Four factors are selected as the main factors affecting Young’s modulus of soft composites: the amount of MRF, the channel pattern, shore hardness and carbonyl iron particle (CIP) concentration of the MRE layer. Five specimens are manufactured to meet the investigation of four factors. Prior to testing, the scanning electron microscopy (SEM) image is taken to check the uniform dispersion of the carbonyl iron particle (CIP) concentration of the MRE layer, and a magnetic circuit is constructed to generate the effective magnetic field to the specimen fixed at the universal tensile test machine. The force–displacement curve is directly measured from the machine and converted to the stress–strain relationship. Thereafter, the Young’s modulus is determined from this curve by performing linear regression analysis with respect to the considered factors. The tunability of the Young’s moduli of the specimens is calculated based on the experimental results in terms of two performance indicators: the relative percentage difference of Young’s modulus according to the magnetic field, and the normalized index independent of the zero-field modulus. In the case of the relative percentage difference, the specimens without MRF are the smallest, and the ones with the highest CIP concentration are the largest. As a result of comparing the normalized index of each factor, the change in shore hardness and channel pattern have little effect on the tunability of Young’s moduli, and the amount of MRF injected and CIP concentration of MRE have a large effect. The results of this study are expected to provide basic guidelines for fabricating soft composites whose field-dependent Young’s moduli can be tuned by several factors with different effects.

## 1. Introduction

It is well known that smart fluids, such as electrorheological fluids (ERFs) and magnetorheological fluids (MRFs), are highly effective for various applications, including automotive damper and vibration control of flexible structures. ERFs consist of fine nonconducting particles and carrier liquids, and their apparent viscosity heavily depends on the electric field that is applied to the fluid domain. Thus, an ERF is generally utilized as a core layer of sandwich structures (beam, shell, and plate) to increase the damping property or alternate the modal characteristics, such as the natural frequency, of the original structures [1,2,3]. However, a very high electric field (1–3 kV/mm) is required to achieve a successful performance change. This drawback stimulated a similar research theme, utilizing an MRF, where low current and low voltage were required to achieve a high-performance effect. As is well known, an MRF comprises iron particles and a carrier liquid to exhibit rheological characteristics that are similar to those of an ERF, as a function of the magnetic field. Because an MRF can produce a significantly higher yield force than an ERF, it is widely applied in many fields, such as in the control of unwanted vibrations in flexible structures. In this case, the MRF, which was sandwiched in outer structures (or skin layers), functioned as the damping core (or constraint layer), whose characteristics depended on the magnetic field that was applied to the fluid domain [4,5,6]. Generally, aluminum, carbon graphite, or steel, which are rigid, are employed as the skin layer. Thus, the structure, incorporating the ERF and MRF, is mostly employed only as the vibration-control beams and plates because it cannot be employed for smart, stretchable, or wearable devices owing to its rigidity. For application in the production of various materials, such as bio-health patches and wearable haptic gloves, and adequate softness of the structure was required. Thus, several scholars have utilized different materials as alternatives to the rigid structure to fabricate smart soft composite structures. Shan et al. [7] proposed a tunable composite, utilizing polydimethylsiloxane (PDMS) elastomer with a low-melting-point metal solder and evaluated the stress–strain behaviors. They confirmed that the Young’s modulus of the soft composite was tuned by four orders of magnitude with respect to electronic activation. Recently, a smart composite, for soft robots, was introduced and discretized with localized geometric patterns [8]. The authors exhibited the stiffness controllability of the proposed soft composite by implementing a shape-memory alloy (SMA) at different temperatures. In the development of soft robotics, the accurate control of the stiffness (or compliance) of the soft composites was crucial [9], and the corresponding control strategies were realized and employed to achieve high performance through the open- or closed-loop control algorithms [10]. The reversible rigidity control was also demonstrated with laminar composites that contained a shape-memory polymer [11], and the phase-changing metal alloy was effectively utilized to reversibly tune the stiffness of the elastomer composites [12]. A method of employing a magnetorheological elastomer (MRE), which is a type of rubber material that was embedded in the magnetic-responsive iron particles [13] as a smart soft composite, has also emerged. Because an MRE is made of rubber, its softness is enough to form various shapes, thus generating a significant change in curvatures. Because of the rheological properties of the MRE, its mechanical properties can be altered by the magnetic field, which is commonly referred to as the tunability of Young’s moduli. The tunability of the Young’s moduli of an MRE was analyzed theoretically and experimentally [14,15,16], and the effectiveness of MRE-based structures was validated in several works [17,18,19]. However, unlike an MRF, an MRE is a solid in which the field-dependent iron particles are restricted in the matrix. Because of this restriction, many studies have been conducted to increase the tunability of the Young’s moduli of the MRE. A hybrid matrix [20,21] and PDMS [22] were utilized as the matrix, and reinforcing fillers [23,24] were added to the matrix to fabricate an MRE with a high tunability of Young’s moduli.

Recently, as a method of increasing the tunability of the Young’s moduli of MRE, a soft composite that used a different method from that of adding an additive to the MRE matrix was suggested. The suggested method introduced in [25] sought to fabricate soft composite consisted of two magnetic-responsive materials—an MRE as the skin layer and MRFs as the cores, which filled the void channels of the lower skin layer. The soft structure was investigated as the proof-of-concept by presenting one preliminary result which clearly shows the field-dependent stiffness. However, the tunability of Young’s moduli of the soft composite is likely to be influenced by many factors, and research on this has not been conducted. Therefore, proceeding from the previous work [25], the effects of factors on tunability of Young’s moduli were investigated, and the significant weight of each factor was identified in this study.

Four factors are selected as the main factors affecting the tunability of the Young’s moduli of soft composites: the amount of MRF, the channel pattern, shore hardness and carbonyl iron particle (CIP) concentration of MRE layer. To compare the effect of each factor, five types of specimens were prepared, and the overall specimen production process was similar to the previous paper [25], but one of the MRF injection amount, the shape of the channel mold, the amount of curing agent entering the MRE layer, and the weight fraction of the CIP entering the MRE layer was varied and applied to each specimen. Prior to the experiment, a magnetic field generator was constructed in the universal tensile test machine. Unlike the previous paper using a neodymium magnet [25], a magnetic circuit using solenoid coils was constructed so that a uniform magnetic field could be applied to the specimen. Fabricated specimens were fixed and tested in a tensile tester, and the directly measured force–displacement curves obtained from the universal tensile machine were converted to the stress–strain curves. Notably, the measured stress–strain relationship was obtained from the MRE skin layer, and the MRF core contained in the channels. Thus, an equivalent Young’s modulus (EYM) was utilized in this paper instead of Young’s modulus. The change in EYM, according to the magnetic field of each specimen, was evaluated using two performance indicators. Finally, the effects of all the factors on the change in EYM were compared. Figure 1 schematically presents application examples of the soft composites; wearable bio-patch, stiffness controllable glove which can be used in virtual work or haptic device, nerve stimulation treatment device, and localized variable stiffness control devices. The results presented in this work are preliminary to explore these applications in the future.

## 2. MRE Preparation

### 2.1. MRE Characteristics

The MRE is one of the materials whose physical properties change according to the magnetic field. When the magnetic field was simulated, the shear modulus of the MRE generally changed in a short time [26,27]. In this study, we verified that the material properties of the MRE–MRF combined structure were changed by the magnetic field. Furthermore, various factors of the MRE production process were considered to demonstrate how the availability of the soft composite changed the EYM property with magnetic stimuli. Figure 2 shows the internal configurations and geometric dimensions of the MRE layer that was utilized in this experiment. The MRE was employed as a container to confine the commercial MRF (MRF 140CG, Lord) and was divided into two pieces. The upper MRE layer was 140 mm long and 0.5 mm thick, and the lower MRE layer was 140 mm long and 1.5 mm thick. The layers were of two types: 25 and 70 mm-wide layers. The lower MRE layers possessed a characteristic pattern and were designed in two forms: the rhombus and rectangular MRE layers. In particular, when the sizes of the specimen were the same but the pattern types were different, the lower MRE layer was designed to possess the same amount of MRF that had been injected (~0.196 mL per unit chamber) and to effectively influence the magnetic field in the center of the length direction. The rhombus and rectangular patterns were selected to compare the arrangement effects on the CIPs that were generated by the magnetic field between the rhombus structure, which was characterized by cusps, and the basic structure of the rectangular structure.

Through the finite element method (FEM) for magnetic field analysis, the effect of the magnetic field was confirmed to be concentrated at the edges and cusps of the pattern, and the average magnetic flux densities acting on the rhombus and rectangle shapes were similar. Additionally, because the magnetic flux was concentrated in the MRF, which exhibited relatively higher permeability than MRE, the MRE walls, separating the chambers, could be neglected in the MR effect. However, because the presented patterns with different cross-section coefficients could not be directly compared, the effects of the patterns were compared experimentally. To evaluate the effect of the shore hardness, which determined the properties of the rubber material, MRE was produced by setting the shore hardness to 30 and 70 and adjusting the ratio of the hardener and silicone rubber. Additionally, to compare the properties of MRE responding to the magnetic field, the proportion of CIP, combined with silicone rubber, was produced, at 40 and 80 wt.% (weight percent), respectively. Therefore, these comparative experiments with the patterns, sizes, shore hardnesses, and CIP content could be a good reference for future studies of soft structures combining the MRE and MRF.

### 2.2. Fabrication of MRE

Figure 3a shows the molds that were produced by computerized numerical control (CNC) machining and employed for the production of the MRE skin layer. Each mold could simultaneously form a lower MRE layer with channels and an upper MRE layer. Figure 3b illustrates the production process of the samples utilized in this experiment. To adjust the shore hardness of the MRE to 30 and 70, the rubber materials containing a hardener, an activator, a silicone rubber, and silicone oil were appropriately combined through repeated experiments. To manufacture MRE with 40 and 80 wt.% of CIP, CIP and the rubber materials were mixed, and thereafter, they were rolled to produce a total of nine rubber sheets with MR particles. After 24 h of aging, an isotropic rubber containing CIP was produced through vulcanization and curing of the compression chamber to achieve elasticity and durability without the MR effect. Next, the MRE shape was manufactured through a press process under the condition that heat was applied to the mold and the thermal deformation to MRE did not occur.

The scanning electron microscopy (SEM) images clearly reveal that CIP was evenly distributed in the mixing process (Figure 4a). An analysis of the SEM images (magnification of 300) of the cross-section of any four points, as shown in Figure 4b, confirmed that CIPs were not concentrated at a specific point of the MRE but were randomly spread across all the points. Consequently, the MRE layer could be regarded as an isotropic MRE in which CIP should be randomly distributed throughout [28].

## 3. Fabrication of the Soft Composite

When attempting to achieve the uniform properties of the soft composites, several steps of bonding were emphasized. Firstly, through the pressing process, the MRE layers of different sizes, pattern shapes, CIP contents, and shore hardnesses were obtained (Figure 5a). Thereafter, the lower layers possessing two intaglio patterns were fixed at a designated position on the table. Secondly, through the dispenser process, MRF was quantitatively injected by one dispenser into the void chamber of the produced lower MRE skin. Lastly, the other dispenser containing the adhesive with a Young’s modulus similar to that for MRE is employed to bond the upper and lower layers, as shown in Figure 3b. Consequently, the structure was symmetrical, and the soft composite combined the MRE and filled MRF. Figure 5b shows the soft composite, which was thin and flexible enough to be bent by hands. Table 1 presents the results (i.e., the mean and standard deviation) obtained from weighing the fabricated specimens. As emphasized in this paper, SEM was performed to observe the interaction between MRE and MRF. Because the SEM observation conditions were limited to the surface of the specimen, the observation of the sample in a magnetic field required five steps during the SEM imaging, as shown in Figure 6a. In the sample-cutting step, MRE was cut into very thin films (~0.2 mm). To fix the membrane, it was attached to a cylindrical specimen holder made of magnetic material with carbon tape. Next, MRF was sandwiched in MREs with a thickness of 0.2 mm. Thereafter, to evaluate the states in the magnetic field, the specimen holder was placed on a permanent magnet inside a non-magnetic square column. At that time, it was sufficiently magnetized for 24 h in the desired magnetic-field direction. The drying step was thereafter required for the specimen to enter the vacuum chamber. The baking process lasted 30 min at 40 °C. As a final pre-treatment process, a platinum-coating step was performed for 120 s to prevent the charge-up phenomena in which the electrons inside and outside the specimen were collected on the sample surface. Next, the samples and the holder were placed in a vacuum chamber and photographed with a scanning electron microscope. Figure 6b shows the sample before magnetization and platinum coating, and it confirms the formation of the CIP chains at the soft composite interface.

## 4. Experimental Setup

To measure the change in EYM of the soft composites according to the magnetic field, an experimental setup (Figure 7) was established. The experimental setup consisted of a tensile test machine and a magnetic field generator. The tensile test machine (KDPI-205 Series, capacity: 1 kN, KD PRECISION Co., Seoul, Korea) tensioned the specimen that was fixed to the chuck with a variable-speed electric motor and a screw bar. Furthermore, the corresponding tensile force was measured from the upper load cell. The measured values were collected and stored in real time with a data logger. The magnetic-field generator was employed to generate two uniform magnetic fields in the x-direction by placing two solenoids that induced a magnetic field by an electric current on both sides of the specimen. The solenoid is a structure wherein a copper wire is wound around a magnetic yoke. The solenoid coil is manufactured by winding AWG27 standard shielded copper wire on 1008 steel yoke, and the coil is wound with an outer diameter of 80 mm and an inner diameter of 45 mm to have a total of 7000 turns. The resistance of the solenoid is 225 ohms, and it takes up to 0.4 A when a voltage of 90 V is applied. A current is supplied from a power source. At that time, the current was applied in each solenoid in opposite directions to induce a uniform magnetic field between the solenoids in the x-direction. A spacer was inserted between the solenoids to create a space of 10 mm, which prevented the solenoids from coming into contact with each other and prevented the specimen from coming into contact with the solenoids. Furthermore, an aluminum jig was manufactured to position the central axis of the solenoids in the middle of the specimen, and the jig with the solenoid was successfully fixed to the bottom of the tensile test machine. When the maximum allowable current of the solenoid, 0.4 A, was applied to the magnetic field generator, the average magnetic-flux density measured by a gauss meter (F.W. BELL 5100 series, MEGGITT, Dorset, UK) was 0.4 T.

Five groups of specimens were selected to compare the effects of shore hardness, pattern shape, CIP concentration, and MRF amount on the change in EYM. The adopted specimen groups were 140 × 25 × 2, shore hardness of 30, CIP of 40 wt.%, and a rectangular pattern type, which was also the reference specimen group; 140 × 25 × 2, shore hardness of 70, CIP of 40 wt.%, and a rectangular pattern type that differed only in hardness from the reference specimen group; 140 × 25 × 2, shore hardness of 30, CIP of 40 wt.%, and a rhombus pattern type with only a different pattern shape; 140 × 25 × 2, shore hardness of 30, CIP of 80 wt.%, and a rectangular pattern type that differed only in CIP concentration; and finally, a group of specimens without MRF, under the same conditions as those of the reference specimen groups. Each specimen group was numbered as follows for easy identification:

Group 1 (Reference): 140 × 25 × 2, shore hardness of 30, CIP of 40 wt.%, and a rectangular pattern;

Group 2: 140 × 25 × 2, shore hardness of 70, CIP of 40 wt.%, and a rectangular pattern;

Group 3: 140 × 25 × 2, shore hardness of 30, CIP of 40 wt.%, and a rhombus pattern;

Group 4: 140 × 25 × 2, shore hardness of 30, CIP of 80 wt.%, and a rectangular pattern;

Group 5: 140 × 25 × 2, shore hardness of 30, CIP of 40 wt.%, and a rectangular pattern without MRF.

Tensile experiments were conducted for the selected specimen groups with the following conditions and procedures: the tensile speed was fixed at 500 mm/min based on American Society for Testing and Materials (ASTM) D412 [29], which is a standard for measuring vulcanized rubber specimens featuring a rectangle shape. The selected tensile length was 6.75 mm, which corresponded to a 5% elongation. In order to prevent stress concentration at the clamp when mounting the specimen in the tensile test machine, specimens were end-tabbed using silicone rubber [30]. A total number of 25 specimens (five specimens for each group) were tested, and each specimen was tested ten times without a magnetic field and ten times with an applied magnetic field of 0.4 T. The tensile length and force obtained through the experiment were converted into strain and stress, respectively, using the stress–strain calculation method of a straight specimen with a constant cross section in test method A of ASTM D412-16 [29]. Preliminary experiments were conducted to confirm intra-specimen repeatability. Figure 8 indicates that five specimens of the same group exhibited almost the same results when stretched by 20 mm both with and without a magnetic field. Furthermore, pre-breaking experiments were carried out, and the rupture that occurred in the middle of the specimen confirms that there was no stress concentration at the clamp.

## 5. Results and Discussions

The stress–strain behaviors of the selected soft-composite specimen groups were determined, employing the previously constructed experimental equipment. The stress–strain curve was expressed as an error plot containing the minimum, maximum and median values of all measured data points. From all the obtained stress–strain curves, EYM, in the absence (zero-field EYM) and presence (field-induced EYM) of a magnetic field, was estimated for each specimen by performing a linear regression analysis. Two performance indicators were employed to analyze the results. The first was the general evaluation index, $PD_{EYM,n}$, which represents the relative percentage difference of EYM according to the magnetic field of the n^th^ group, calculated by Equation (1):(1)$$PD_{EYM,n}=\frac{∆EYM_{n}}{EYM_{0T,n}}\times 100=\frac{EYM_{0.4T,n}-EYM_{0T,n}}{EYM_{0T,n}}\times 100$$
where ∆EYM, $EYM_{0T}$, and $EYM_{0.4T}$ denote the variations in EYM, i.e., the zero-field EYM and field-induced EYM of the n^th^ group, respectively. The second was the normalized index ($∆EYM_{0.4T}/∆EYM_{0T}$), which represents how much each factor altered the field-induced modulus contrary to changing the zero-field modulus. It is calculated by Equation (2):(2)$${∆EYM_{0.4T}/∆EYM_{0T}}_{n}=\frac{\left|EYM_{0.4T,1}-EYM_{0.4T,n}\right|}{\left|EYM_{0T,1}-EYM_{0T,n}\right|}$$

Through the calculation of $∆EYM_{0.4T}/∆EYM_{0}$, the influence of each factor could be normalized based on Group 1.

In Figure 9, the tensile test results for Groups 1 (reference group) and 2 are compared. The figure shows the effect of shore hardness on EYM change with a magnetic field. First, $EYM_{0T,1}$ and $EYM_{0T,2}$ were 0.89 and 2.61 MPa, respectively. $EYM_{0T,2}$ with a high shore hardness exceeded $EYM_{0T,1}$, and these results agree with the existing investigation, wherein the relationship between the hardness of silicone rubber and Young’s modulus was experimentally analyzed [31]. Further, $EYM_{0.4T,1}$ and $EYM_{0.4T,2}$ were 2.40 and 4.18 MPa, respectively. Additionally, $∆EYM_{1}$ and $∆EYM_{2}$, 1.51 and 1.57 MPa, respectively, were almost the same, but $PD_{EYM,1}$ and $PD_{EYM,2}$ were 169.34% and 59.97%, respectively. Group 2 exhibited a significantly lower relative percentage increase because its zero-field modulus exceeded that of Group 1, although the variations in EYM were similar. Furthermore, ${∆EYM_{0.4T}/∆EYM_{0T}}_{2}$ was 1.032, indicating that only the field-induced modulus changed as much as the zero-field modulus changed when shifting from Group 1 to 2. Figure 10 shows the effect of channel pattern on the change in EYM according to the magnetic field, based on the tensile tests of Groups 1 and 3. $EYM_{0T,3}$ was 1.12 MPa, which slightly exceeded $EYM_{0T,1}$. $EYM_{0.4T,3}$ and $∆EYM_{3}$ were 2.67 and 1.55 MPa, respectively. These values are almost the same as those of $∆EYM_{1}$ and $∆EYM_{2}$. Additionally, the calculated $PD_{EYM,3}$ was 137.95%, which was relatively low compared to $PD_{EYM,1}$. This result was obtained because the zero-field modulus of Group 3 slightly exceeded that of Group 1 as the section modulus of the specimen changed, although each variation of EYM barely changed. ${∆EYM_{0.4T}/∆EYM_{0}}_{3}$ was 1.16, which was almost the same as the value obtained for ${∆EYM_{0.4T}/∆EYM_{0}}_{2}$. Figure 11 indicates the effect of CIP concentration on the change in EYM according to the magnetic field by comparing the tensile test results of Groups 1 and 4. $EYM_{0T,4}$ was 1.31 MPa, which slightly exceeded $EYM_{0T,1}$. The increase in the zero-field EYM was due to a perturbation of the stress and strain by suspended particles owing to an increase in the CIP concentration, and this perturbation increased the elastic energy [32]. $EYM_{0.4T,4}$ was 4.16 MPa, and $∆EYM_{4}$ was 2.84 MPa, representing a significant increase, compared to the previous results. Additionally, $PD_{EYM,4}$ was 216.36%, which remarkably exceeded $PD_{EYM,1}$. This improvement occurred because the variation in EYM of Group 4 significantly exceeded that of Group 1, although the zero-field modulus of Group 3 exceeded that of Group 1. ${∆EYM_{0.4T}/∆EYM_{0}}_{4}$ was 4.16. This indicated that shifting from Group 1 to 4 correspondingly changed the field-induced modulus to approximately four times the change from the zero-field modulus. Figure 12 compares the tensile test results of Groups 1 and 5 and indicates the effect of the MRF amount on the EYM change, according to the magnetic field. $EYM_{0T,5}$ was 1.08 MPa, almost the same as $EYM_{0T,1}$. It could be deduced that the inserted MRF hardly changed the elastic energy. $EYM_{0.4T,5}$ and $∆EYM_{5}$ were 1.47 and 0.39 MPa, respectively, which were significantly lower than the values obtained in the previous results. $PD_{EYM,5}$ was 36.49%, which was lower than the value of $PD_{EYM,1}$. This decrease was because the zero-field modulus of Group 4 was almost the same for Group 1, even though MRF was not injected, but the variation in EYM for Group 5 was significantly lower than that of Group 1. ${∆EYM_{0.4T}/∆EYM_{0}}_{5}$ was 4.96, indicating that the field-induced modulus had changed to approximately five times the change in the zero-field modulus because of the shift from Group 1 to 5.

To summarize, all the groups exhibited an increase in EYM when a magnetic field was applied. By varying each factor, $PD_{EYM}$ could be changed from a minimum of 36.49% to a maximum of 216.36%. Further, $∆EYM_{0.4T}/∆EYM_{0}$ was large (in the order of Groups 5, 4, 3, and 2). Accordingly, the order of the normalized index is the amount of MRF, the CIP concentration of the MRE skin layer, the channel pattern, and the shore hardness. The radar chart, as shown in Figure 13, summarizes the effect of each parameter, and the expressed data are based on the calculated normalized index of each parameter. In the cases of Groups 2 and 3, the calculated values of $∆EYM_{0.4T}/∆EYM_{0}$ were ~1, indicating that the two factors were not key to the additional change in the field-induced modulus. That is, when comparing the reference group and the Group 2, the CIP trapped in the MRE may have different boundary conditions depending on the hardness of the rubber wall. The experimental results show that the MRE’s shore hardness increasing the binding of CIP to the MRE skin layer did not affect the cohesiveness of particle chains and was not a dominant factor in Young’s moduli. In addition, when observing Group 3 during the tensile test, two vertex angles located in the direction of the tensile axis of the MRE with rhombus shape are narrowed from a right angle to an acute angle according to the Poisson’s ratio. The MRF in the cups of the rhombus is then subjected to compressive forces so that the MRF can be operated in squeeze mode. However, it is observed that the tensile speed is too slow and the shrinkage of the MRF chamber is small, so that it does not generate a significant resistance to compression deformation.

On the other hand, the values of $∆EYM_{0.4T}/∆EYM_{0}$ in Groups 5 and 4 significantly exceeded that in Group 1. The cause of the increase in normalized index by increasing the CIP concentration of MRE layer can be speculated from the interaction energy between the CIPs. In the presence of the magnetic field, the interaction energy between particles is proportional to the dipole moment and inversely proportional to the permeability of the medium and the distance between neighboring particles [33]. As the CIP concentration increases, the distance between neighboring particles per volume decreases, which causes an increase in magnetic interaction energy. As a result, it can be seen that the magnetic cohesion of the neighboring particles increases, resulting in the increment of the MR effect.

In addition, the main cause of the increase in normalized index due to the increase in the amount of injection of MRF is not only the interaction energy but also the mobility of particles. As the number of CIPs per volume increases due to the injection of MRF, the distance between particles becomes small, and the CIP included in MRF has a relatively higher degree of freedom than the CIP included in MRE. Thus, it can move and form a chain. The formation of this chain can be observed in SEM images at the interface of MRE and MRF (refer to Figure 6b). This indicates that the distance between the particles included in MRE and MRF is very important. In other words, if the distance becomes smaller, the MR effect is increased by further increasing the magnetic cohesion between the CIPs included in both MRE and MRF.

However, the limitations should be noted. In order to clearly distinguish the effect of each parameter from the influence of the deviation of the specimen, an approach to apply a massive magnetic field of 0.4 T was attempted. Since a large magnetic field of 0.4 T is difficult to implement in compact applications adjacent to the human body, the soft composite should achieve a stiffness change in a small magnetic field in practice. To achieve a large stiffness change in a small magnetic field, soft composites need to be optimized based on the normalized index discussed in this work and the behavior of the optimized soft composites should be explored in the small magnetic field.

## 6. Conclusions

In this study, soft composites, consisting of MRE and MRF, were fabricated to investigate the field dependent tunable EYM. In particular, to identify the most significant factors that could affect the change in EYM, several soft composite samples were fabricated in consideration of several parameters: different magnetic field intensities, different MRF amounts, different shore stiffnesses of MREs, different CIP concentrations of MREs, and different patterns of the void channels of the lower skin layer, which was to be filled with MRF. The CIP distribution of the produced MRE and the interface of MRE and MRF were observed through SEM imaging. The universal tensile test machine and magnetic circuit for the uniform magnetic field were configured, and the force–displacement behaviors of the specimens with and without the magnetic field were determined. The overall experimental data were transformed into stress–strain relationship data to estimate EYM. By varying some factors, the relative percentage difference of EYM, according to the magnetic field, could increase up to 216.36%. As a result of normalizing the effects of each factor, based on the reference group, the MRF amount had the greatest effect on the change in EYM with the magnetic field, and the shore hardness of MRE was barely related to the change in EYM with the magnetic field. More specifically, the calculated normalized index of the MRF amount, the CIP concentration of MRE, the channel pattern, and the shore hardness of MRE were 4.96, 4.16, 1.16, and 1.03, respectively. Finally, future research should be devoted to figuring out the behavior of soft composites in the low magnetic field region in order to effectively realize the soft composite in various applications.

## Figures and Tables

**Figure 1 materials-13-03378-f001:**
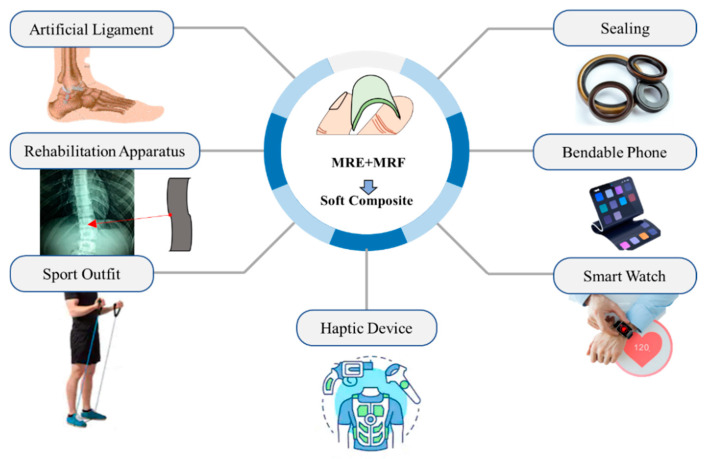
Various applications of soft composites fabricated from magnetic-responsive materials.

**Figure 2 materials-13-03378-f002:**
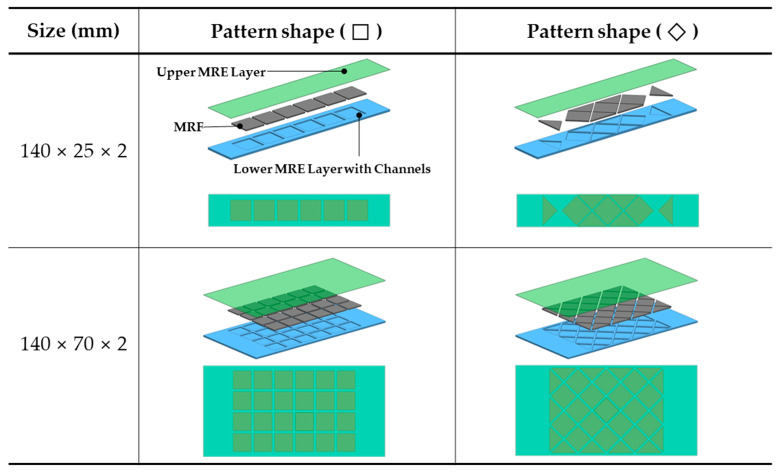
Schematic configuration of the soft composites fabricated from magnetorheological elastomer (MRE) layers and magnetorheological fluid (MRF).

**Figure 3 materials-13-03378-f003:**
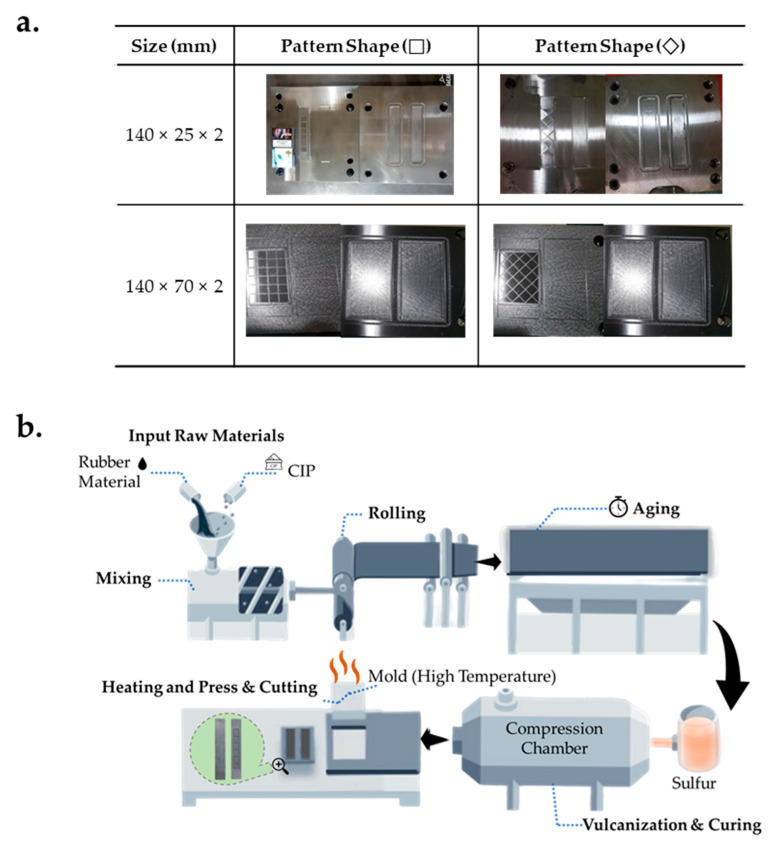
Procedures for fabricating the upper and lower MRE layers: (**a**) photos of mold; (**b**) fabrication phases.

**Figure 4 materials-13-03378-f004:**
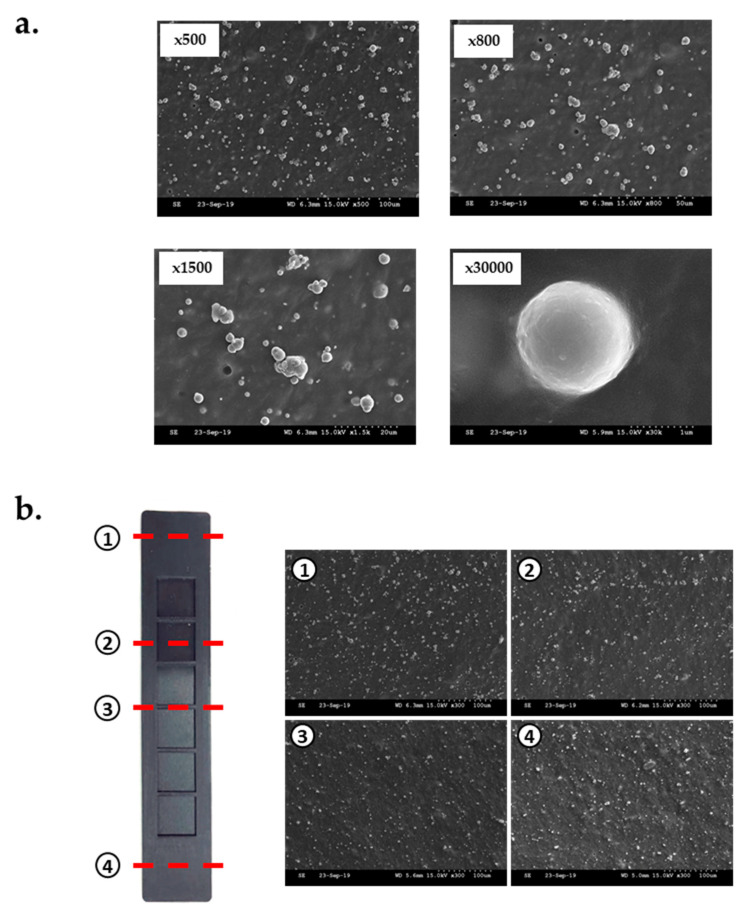
Scanning electron microscopy (SEM) images of the MRE layer: (**a**) with different magnifications and (**b**) uniform distribution of carbonyl iron particle (CIP).

**Figure 5 materials-13-03378-f005:**
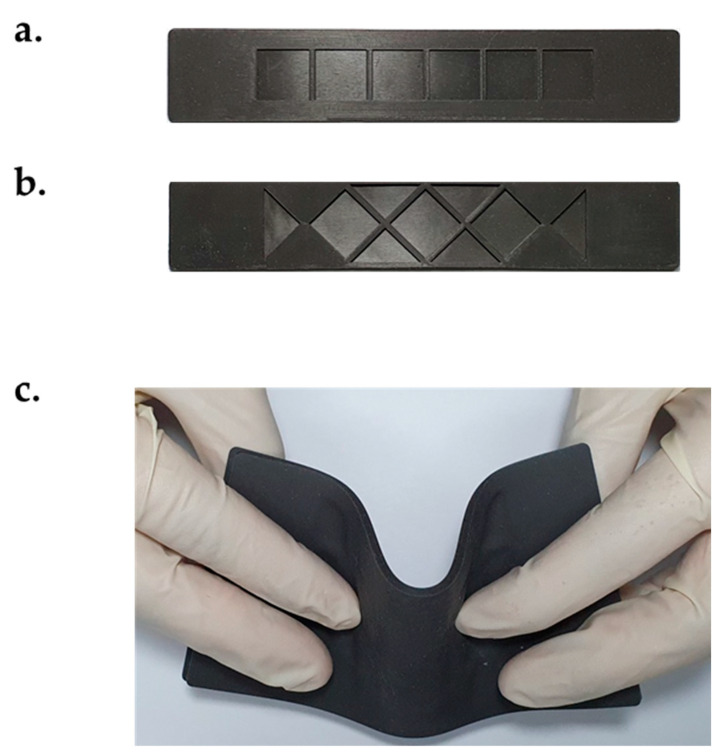
Pictures of the manufactured soft composites: (**a**) rectangular pattern, (**b**) rhombus pattern, and (**c**) bent shape.

**Figure 6 materials-13-03378-f006:**
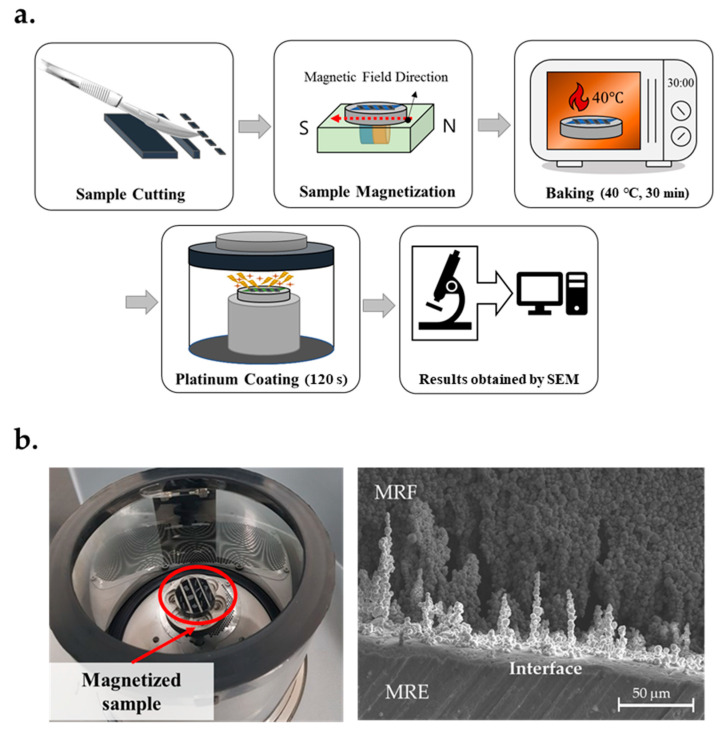
Magnetic-responsive microsized CIP steps of the soft composites: (**a**) SEM procedure and (**b**) magnetized sample and SEM image.

**Figure 7 materials-13-03378-f007:**
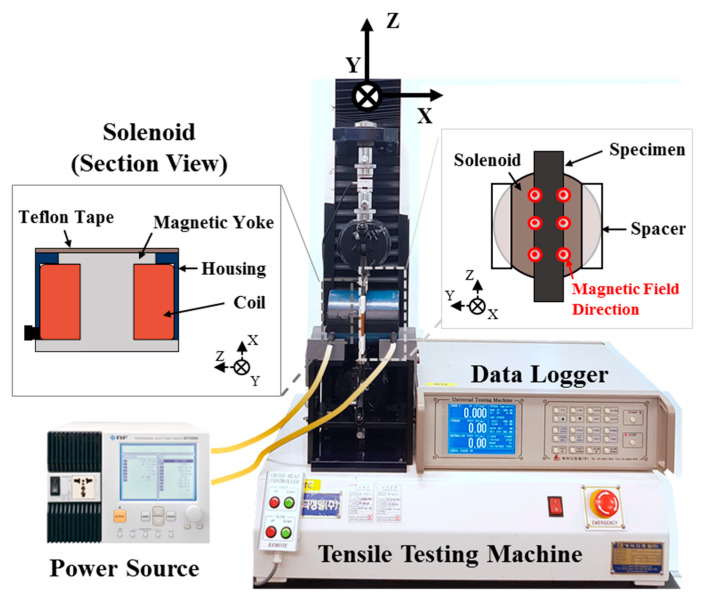
Tensile testing machine with the magnetic core structure.

**Figure 8 materials-13-03378-f008:**
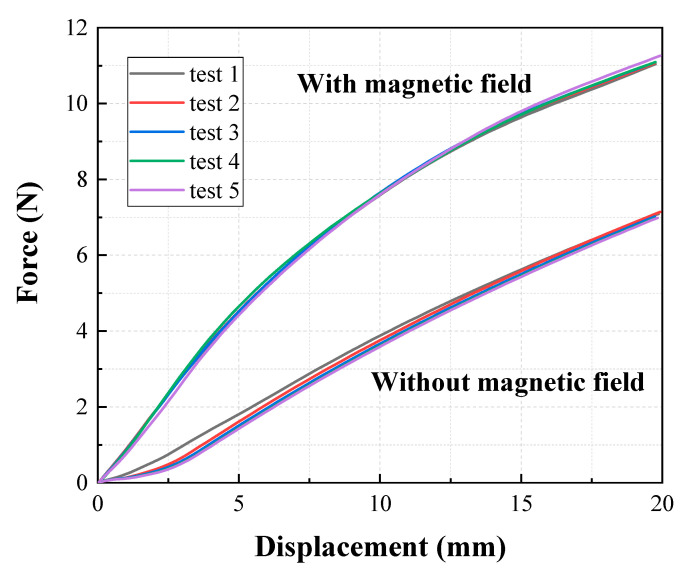
Intra-specimen repeatability test results of five samples (rectangular) under the same conditions.

**Figure 9 materials-13-03378-f009:**
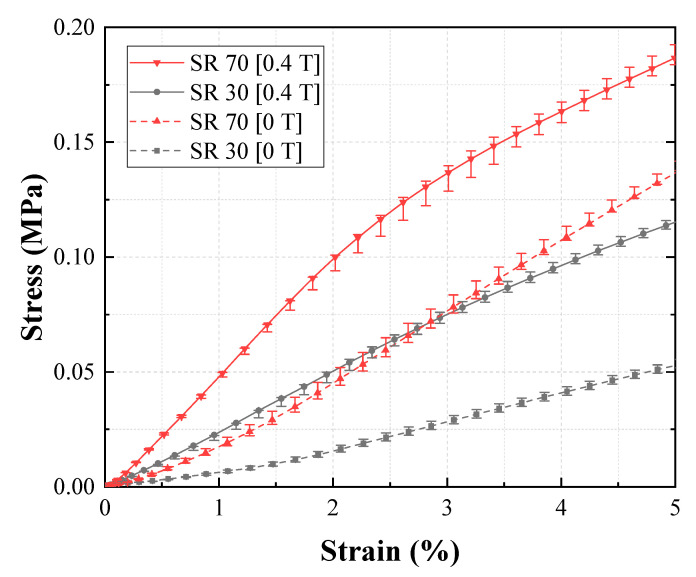
Effect of the shore hardness of the MRE layers on Young’s modulus.

**Figure 10 materials-13-03378-f010:**
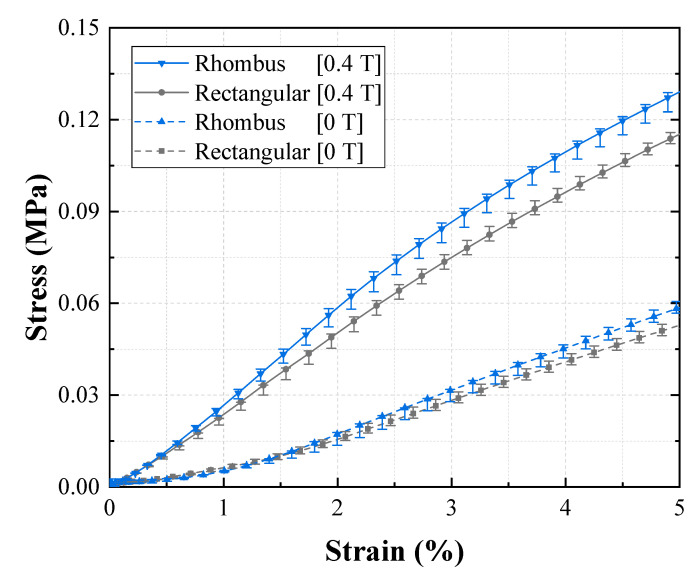
Effect of the channel pattern of the lower MRE layer on Young’s modulus.

**Figure 11 materials-13-03378-f011:**
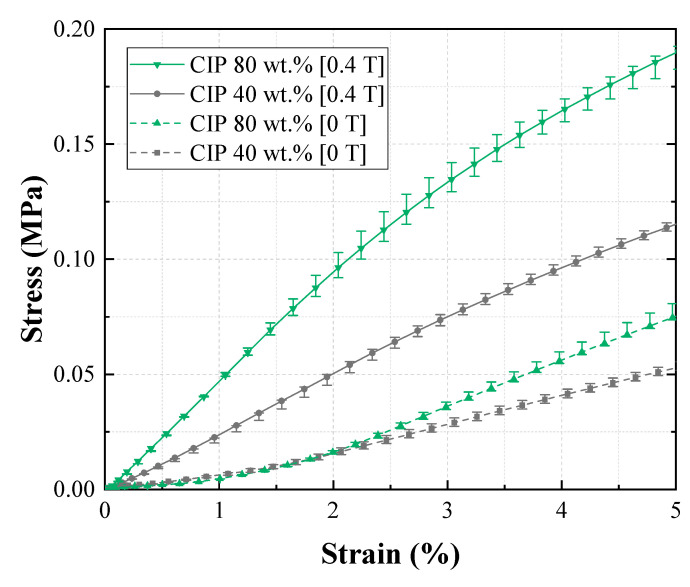
Effect of the CIP concentration of the MRE layers on Young’s modulus.

**Figure 12 materials-13-03378-f012:**
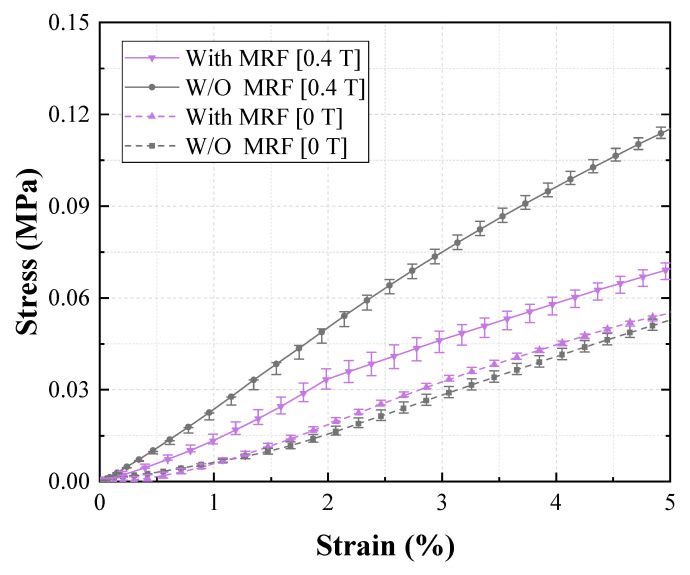
Effect of MRF contained in the channels of the lower MRE layer on Young’s modulus.

**Figure 13 materials-13-03378-f013:**
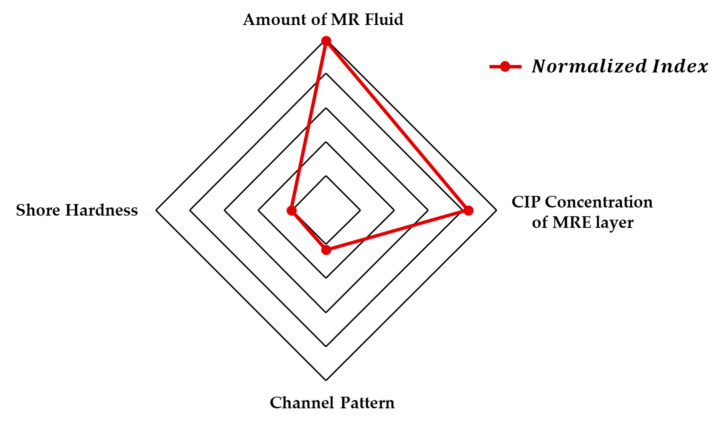
Radar chart of each parameter.

**Table 1 materials-13-03378-t001:** Mean and standard deviation of the weight per specimen.

Specimen No.	Shore Hardness	Pattern	Concentration of CIP (wt.%)	MRF Injection	Weight of Specimens (g)						
					#1	#2	#3	#4	#5	Mean	SD
1 (Reference)	30	■	40	Y	14.8	15.6	14.8	15.2	14.9	15.1	0.344
2	70	■	40	Y	16.2	16.4	16.1	16.6	17.1	16.5	0.396
3	30	◆	40	Y	18.3	17.9	17.9	17.9	17.9	18.0	0.179
4	30	■	80	Y	25.4	25.2	25.3	25.2	25.0	25.2	0.148
5	30	■	40	N	11.4	10.6	10.8	10.7	10.6	10.8	0.335

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
