# Peer review of "Tunable Young’s Moduli of Soft Composites Fabricated from Magnetorheological Materials Containing Microsized Iron Particles"

_materials, 2020, doi:10.3390/ma13153378_

Round 1
Reviewer 1 Report
The paper, reports results of research focused on the influence of the magnetic field on the value of the Young module. In the paper, the authors present a complex description of the research methodology. The description of the results is at the fine level.
However, few items are to be addressed before publication. The manuscript can be improved by considering the following:
- quality of figures 1, 2 4a could be better, I suggest clearing a background
- in the description of sample dimensions I suggest to use “x” symbol, not a “*”, for example, 140x25x2
- in chapter Results and Discussions authors use the wrong symbol of the unit of stresses. I suggest change a "Mpa" to the MPa.
Reviewer 2 Report
Th
The peer-reviewed article is a continuation and development of the methods and approaches to fabricate very interesting soft composite structure consisted of two magnetic-responsive materials which was proposed by the authors in a recent publication [25]. This fact does not make the article less significant. A well-planned thorough study of the Young's modulus in such composites and its dependence on the external magnetic field constitute the main content of the paper. Many factors affecting this dependence have been taken into account and optimal sets of parameters have been determined that allow effective control of the mechanical properties of the composite. The work makes a very good impression, it is written and illustrated with great respect to readers and the Journal and, of course, is important for the development of nanotechnology. I recommend to accept the work to publication with very small corrections.
Page 2, lines 75, 76 – It is desirable to change the wording “In this work, a new soft composite structure … was proposed”, since this structure was proposed earlier in [25].
Reference [25] should be supplemented by volume and article numbers.
Authors should carefully check for errors such as time inconsistencies, singular and plurals, as well as typos.
e peer-reviewed article is a continuation and development of the methods
Reviewer 3 Report
In this work, the authors investigate soft materials filled with MR fluids that exhibit different modulus upon applying fields. The authors test a few select material texturings. The dramatic stiffness response under the application of magnetic fields is very cool. There are some things the authors need to take care of in my mind which precludes the paper from being deemed acceptable.
- Figure 8-11 are the bulk of the technical work, but it looks like the authors didn't gather sufficient samples (N=1???). The authors are reporting a single result for each class of sample. More samples are required to identify the natural variability of the processing method and to determine what differences could be due to different designs versus processing inconsistencies. Since this is the bulk of the study, the reviewer sees this as a major flaw.
- Figure 7 shows Newton versus Displacement, while Figure 8-11 show MPa vs strain. The sample geometry the authors are using is not a standard dogbone and there will be stress concentrations at the clamps. How are the authors accurately determining stress/strain? How much does this stress concentration affect their data?
- 0.4T is a very large magnetic flux density. The applications that the authors discuss are sports outfits, smart devices, etc... A magnetic flux density of 0.4T could not be implemented into these applications. This begs the question of how relevant this study is. The authors should additionally measure the stiffness differences at a much more modest magnetic field that is more reasonable for the applications. This allows the readership to judge this approach in light of the promise.
- The materials used are incredibly soft, at 100 kPa modulus. The reviewer assumes that stiffer materials would have the elastic energy to completely mute these magnetic field based stiffness changes. The authors should calculate and show the limitations of the elastic properties of the base material to clearly show in what range the materials have to be...(very soft).
- The radar chart needs more clarification. I guess the authors just want to make the point that MR amount and concentration play the predominant role. Is the radar plot the best way to show that? What is the data based on?
- The reviewer suggests that the authors lead with the discussion of applications, putting into the intro, to motivate the work.
- Section 5 is Results and Discussion, but there is no real discussion, just a list of results. The reviewer suggests that not enough systematic studies were done to have an enriched discussion.
- The reviewer, who has worked with solenoids a fair amount, has no idea how the authors would generate 0.4T with only 0.4 A. More details about how this massive field with minimal current is requested to understand the setup.
Round 2
Reviewer 3 Report
The authors have addressed many of my concerns. Unfortunately a new concern has arisen in my re-review. I read the authors' previous publication (which I missed in the first review) and I was astonished how this system has been before reported in this publication:
Materials 2020, 13(4), 953; https://doi.org/10.3390/ma13040953
The authors indeed cited this and I should have read the background article to make my first review. The entire concept and system is published in this work.
I was misled because the authors discuss their work as if completely novel. For example, their first sentence is "A new soft composite, fabricated from two different magnetic-responsive materials, namely magnetorheological elastomer (MRE) and magnetorheological fluid (MRF), is proposed to investigate the tuning capability of Young’s modulus".
In light of the previous publication, this soft composite is neither new or now proposed. Thus, I find it hard not to call this a "very misleading" statement.
The key difference between the two papers is that now the authors rotate the square pattern and call it rhombus and look at changing concentrations and amounts of MR. (Aside: Though technically a square is a rhombus.) The difference between the square and "rhombus" patterns are relatively small and not explained in any meaningful way. Therefore, there is a lot less substance to this paper than I had previously imagined.
- The authors need to remove misleading statements about these materials and methods being new or novel.
- The authors need to refocus the impact statements and the abstract that this paper looks at a few parameter dependencies that weren't fully discussed in the first paper.
- The new findings of this paper can be summarized that the two chosen geometries are similar and that higher amounts and concentrations of MR makes a stronger impact. Though the higher amount and concentration leads to a bigger response is obvious - the authors still do not discuss this with enough detail - since it's now one of the "new" things. Also, the role of geometry and why the simple ones considered don't change things too much (though there are some differences!) are not sufficiently explained.
Currently with the misleading statements and the lack of scientific discussion about what is actually new, I recommend rejection.
Round 3
Reviewer 3 Report
I appreciate the authors reworking their language to clearly present this work as an extension of their previous work. I no longer have ethical concerns with the manuscript.
I feel the manuscript is now technically sound and showcases some additional studies on their previously developed system.